# Dual-stage Contrastive Learning-enhanced Multi-view Variational Clustering

**Yanxi Liu** [1]   **Yipin Hu** [1]   **Fangxi Liu** [1]   **Yanwei Yu** [2]   **Lei Meng** [3]   **Yongyong Chen** [4]   **Guoqing Chao** [1]

## Abstract

Multi-view clustering aims to obtain a consensus clustering by integrating complementary and consistent information from multiple views. However, two critical challenges still exist in variational methods: (1) view heterogeneity and noise often make fusion unreliable; (2) ambiguous posteriors and misassigned boundary samples impact the clustering performance. To address these issues, we propose Dual-stage Contrastive Learning-enhanced Multi-view Variational Clustering (DCL-MVC), which integrates contrastive learning into both the fusion and representation stages. Firstly, at the fusion stage, we introduce a fusion-then-attention mechanism to capture cross-view interactions and learn view-level attention weights for building a unified and reliable fused representation, and further introduce instance-level contrastive learning to enforce cross-view alignment at the instance level. Secondly, we focus on boundary samples with uncertain posteriors and refine their cluster assignments by using cluster-center contrastive loss to enlarge inter-cluster margins, while leveraging prototypical contrastive learning with a confidence-aware curriculum to promote intra-cluster compactness at the representation stage. Extensive experiments on six real-world datasets demonstrate consistent improvements over strong baselines and validate the contribution of each component.

## 1. Introduction

With the rapid development of sensing and data acquisition technologies, data in real-world applications often come in multiple heterogeneous views, such as images accompanied by textual descriptions, camera data captured from different viewpoints, and time-series signals recorded by different sensors (Zhao et al., 2017; Xu et al., 2013). A central goal of multi-view learning is to maximize inter-view agreement by extracting view-invariant structures, while mitigating view-dependent variations, redundancy, and noise (Zhao et al., 2017). As an important branch of multi-view learning, multi-view clustering (MVC) aims to uncover consistent latent structures shared across views in an unsupervised manner and partition samples into semantically coherent clusters (Yang & Wang, 2018; Kumar et al., 2011). It has been widely used to produce clustering results that benefit downstream tasks such as retrieval, classification, and recommendation (Wang et al., 2021b; Xu et al., 2018; Yang & Wang, 2018).

However, MVC in practical scenarios remains challenging due to severe view heterogeneity and ambiguous cluster boundaries, making it difficult to learn models that are both robust and highly discriminative (Fu et al., 2024). A key challenge lies in reliable cross-view fusion under heterogeneity and noise. Most existing fusion strategies either concatenate view-specific embeddings, perform late fusion (Liu et al., 2021), or adopt adaptive attention-based weighting to estimate view importance (Huang et al., 2023; Wang et al., 2022; Wu et al., 2024). However, view-level aggregation alone does not guarantee instance-level cross-view alignment (Chao et al., 2021; Chuang et al., 2020): when views disagree or contain noise, the fused feature can still be unstable and even biased toward dominant views (Liu et al., 2021; Huang et al., 2023), which undermines clustering reliability. This motivates a fusion mechanism that jointly models cross-view interactions, learns global view-level weights, and explicitly enforces instance-level alignment to obtain a reliable fused signal for clustering.

Another fundamental challenge arises from posterior ambiguity in the latent space, especially near inter-cluster boundaries. When the fused representation is unreliable under view heterogeneity and noise, this unreliability can be amplified by variational inference, leading to overlapping pos-

---

[1] School of Computer Science and Technology, Harbin Institute of Technology, Weihai, China [2] Faculty of Information Science and Engineering, Ocean University of China, Qingdao, China [3] School of Software, Shandong University, Jinan, China [4] School of Computer Science and Technology, Harbin Institute of Technology, Shenzhen, China. Correspondence to: Guoqing Chao <guoqingchao@hit.edu.cn>.

*Proceedings of the 43rd International Conference on Machine Learning*, Seoul, South Korea. PMLR 306, 2026. Copyright 2026 by the author(s).

teriors and unstable cluster assignments for boundary samples. Existing MVC methods, even with VAE-style latent mixture modeling, often incorporate discriminative signals only weakly, making it difficult to separate clusters once posteriors overlap (Chao et al., 2021; Yang & Wang, 2018; Kingma & Welling, 2014; Rezende et al., 2014; Kingma et al., 2019; Xu et al., 2021). Moreover, boundary samples are frequently treated as confident ones or filtered by crude thresholds (Xie et al., 2016a), which may either inject erroneous pseudo-supervision and amplify assignment errors, or discard informative hard cases. Therefore, beyond reliable fusion, a dedicated representation stage is needed to explicitly refine boundary assignments and enhance inter-cluster separability in the latent space.

To address the above issues, this paper, under the framework of variational autoencoders, proposes a deep multi-view clustering method termed Dual-stage Contrastive Learning-enhanced Multi-view Variational Clustering (DCL-MVC), which integrates generative modeling, attention-based fusion, and dual contrastive learning. The main contributions of this work are summarized as follows:

- We propose DCL-MVC, a VAE-based generative multi-view clustering framework with a GMM prior. Observing that unreliable view fusion can be amplified by variational inference and lead to ambiguous posteriors, we introduce a dual-stage formulation to address fusion reliability and posterior uncertainty at different stages, thereby enhancing clustering robustness.

- We introduce a fusion-then-attention strategy that models cross-view interactions and learns view-level importance weights to form a unified representation, and further apply instance-level contrastive alignment to enforce cross-view consistency, stabilizing the fused representation used by the generative posterior.

- We develop a boundary-aware contrastive learning strategy to address boundary samples near inter-cluster transition regions, where cluster-level and prototype-level supervision jointly encourage clearer cluster separation and guide uncertain samples toward more appropriate cluster assignments.

## 2. Related Works

### 2.1. Deep Multi-view Clustering

In recent years, deep multi-view clustering has become the mainstream paradigm, with many studies aiming to learn clustering-friendly shared representations. Fu et al. (Fu et al., 2024) learn a consistent shared subspace via subspace contrastive learning with structural regularization, while Huang et al. (Huang et al., 2023) exploit self-supervised

signals to adaptively weight views and refine common representations but may still be sensitive to fusion noise. For incomplete multi-view data, existing methods leverage cross-view partial alignment and structural completion (Jin et al., 2023) or proxy supervision between paired embeddings (Cai et al., 2024) to handle missingness and imbalance. Yu et al. (Yu et al., 2025) further introduce a mutual-information formulation and couple incomplete-view prediction with contrastive clustering for joint optimization. In federated scenarios, Chao et al. (Chao et al., 2025) propose globally fused graph guidance with pseudo-label refinement to enable privacy-preserving clustering under missing views. Moreover, adversarial and generative frameworks, such as AMvC (Wang et al., 2022), DAMC (Li et al., 2019), and cycle-consistent generation (Wang et al., 2021a), are developed to balance cross-view consistency and diversity for more explicit cluster-structure modeling.

### 2.2. Variational Autoencoders

Variational autoencoders (VAEs) (Kingma et al., 2019) are a class of deep generative models that introduce a prior distribution in the latent space and maximize the evidence lower bound (ELBO) to probabilistically model the data-generating process. For clustering tasks, many works extend the standard VAE by incorporating a Gaussian mixture prior or discrete cluster indicator variables (Jiang et al., 2016; Dilokthanakul et al., 2016; Xu et al., 2024), so that different clusters correspond to distinct latent components and cluster structure can be discovered jointly with generative modeling; Dupont (Dupont, 2018) further designs a disentangled latent space to jointly learn continuous and discrete representations.

In multi-view settings, Xu et al. (Xu et al., 2021) propose a VAE-based multi-view clustering framework that models view-shared variables with an approximately discrete Gumbel–Softmax distribution and view-specific variables with Gaussian distributions, using mutual-information regularization to disentangle them and improve both clustering performance and interpretability. Yin et al. (Yin et al., 2020) learn a shared generative latent representation governed by a Gaussian mixture within a VAE, which captures nonlinear features from each view as well as inter-view correlations and achieves strong performance, yet overlapped posteriors can still occur. For partially aligned multi-view data, Yang et al. introduce the causal multi-view clustering network Cau-MVC (Yang et al., 2025), embedding causal learning and a contrastive regularizer into a VAE architecture to estimate invariant features and sample-wise dependencies, and achieving strong generalization on both fully and partially aligned datasets.

## 2.3. Boundary-aware Clustering

Recent clustering studies have increasingly recognized that pseudo assignments are not equally reliable, and many methods improve clustering quality by treating samples with different confidence levels differently. A common strategy is to emphasize high-confidence sample selection during training. For example, DEC refines cluster assignments through a sharpened target distribution, which effectively strengthens the role of confidently assigned samples (Xie et al., 2016b). Similar ideas also appear in SCAN and SPICE, where reliable pseudo-labels or confident predictions are used to stabilize representation learning and cluster optimization (Van Gansbeke et al., 2020; Niu et al., 2022). Although these methods improve pseudo-supervision quality, they mainly focus on exploiting easy samples, rather than explicitly modeling samples located near cluster boundaries.

Compared with such confidence-based strategies, another line of work pays more direct attention to boundary samples. Some methods improve cluster discrimination at the prototype level by enlarging inter-prototype distances or strengthening sample-to-prototype alignment, which can indirectly reduce ambiguity around cluster transitions (Huang et al., 2022). Other methods attempt to explicitly identify boundary samples according to geometric or neighborhood structure. For instance, ConNR introduces a progressively relaxed neighborhood-refinement strategy to reduce the harmful influence of unreliable neighbors near cluster boundaries (Yu et al., 2023).These studies indicate that distinguishing boundary samples from easy samples is increasingly important for improving clustering quality.

## 3. Proposed Method

In this section, we first introduce the notations for clarity and consistency, and then describe the components of our proposed method.

### 3.1. Problem Formulation

Let $X = \{X^1, \ldots, X^V\}$ denote a multi-view dataset with $V$ views. The data matrix of the $v$-th view is denoted by $X^v \in \mathbb{R}^{N \times d_v}$, where $d_v$ is the dimensionality of this view, $N$ is the number of samples, and $V$ is the number of views. The $v$-th view of the $i$-th sample is denoted by $x_i^v$, thus, $X^v = \{x_i^v\}_{i=1}^N$. The goal of VAE-based multi-view clustering is to learn a shared $D$-dimensional continuous latent representation $z \in \mathbb{R}^D$ from multi-view data and then cluster the $N$ samples in this latent space into $K$ clusters.

### 3.2. Generative Process

Within the VAE-based multi-view clustering framework, we introduce a discrete cluster indicator $c \in \{1, \ldots, K\}$ and

a continuous latent variable $z \in \mathbb{R}^D$ shared across views. The prior assumptions of the model are the same as in the work (Jiang et al., 2016).

First, the discrete clustering variable $c$ follows a categorical prior:

$$p(c = k) = \pi_k, \quad \sum_{k=1}^K \pi_k = 1, \tag{1}$$

where $\pi_k$ denotes the prior weight of the $k$-th cluster. Conditioned on the cluster $c$, the shared latent variable $z$ is assumed to follow a Gaussian distribution associated with cluster $c$:

$$p(z \mid c = k) = \mathcal{N}\big(z \mid \mu_k, \sigma_k^2 \mathbf{I}\big), \tag{2}$$

where $\mu_k$ and $\sigma_k$ are the mean and scale parameters of the $k$-th Gaussian mixture component, encoding the center and shape of each cluster in the latent space. This leads to the overall Gaussian mixture prior:

$$p(z) = \sum_{k=1}^K \pi_k \, \mathcal{N}\big(z \mid \mu_k, \sigma_k^2 \mathbf{I}\big), \tag{3}$$

where $\{\pi_k, \mu_k, \sigma_k\}_{k=1}^K$ are all the parameters that can be learned in an end-to-end fashion during training, without relying on pretraining.

Unlike approaches that introduce independent latent variables for each view, our model uses a single shared latent variable $z$ to generate all views:

$$x^v \sim p_{\theta_v}(x^v \mid z), \tag{4}$$

where $\theta_v$ denotes the parameters of the decoder for the $v$-th view. Under this generative mechanism, the joint probability of the multi-view sample and its latent variables can be written as:

$$p\big(\{x^v\}_{v=1}^V, z, c\big) = p_\theta\big(\{x^v\}_{v=1}^V \mid z\big) \, p(z \mid c) \, p(c). \tag{5}$$

For multi-view clustering, we further assume that the views are conditionally independent given the shared latent representation $z$. Then the joint distribution of the data and latent variables can be factorized as:

$$p\big(\{x^v\}_{v=1}^V, z, c\big) = p(c) \, p(z \mid c) \prod_{v=1}^V p(x^v \mid z). \tag{6}$$

### 3.3. Variational Inference

The objective of the variational inference process is to learn the joint posterior distribution $p(z, c \mid \{x^v\}_{v=1}^V)$. Since the exact posterior is intractable, we adopt variational inference and introduce a variational distribution $q(z, c \mid \{x^v\}_{v=1}^V)$ to

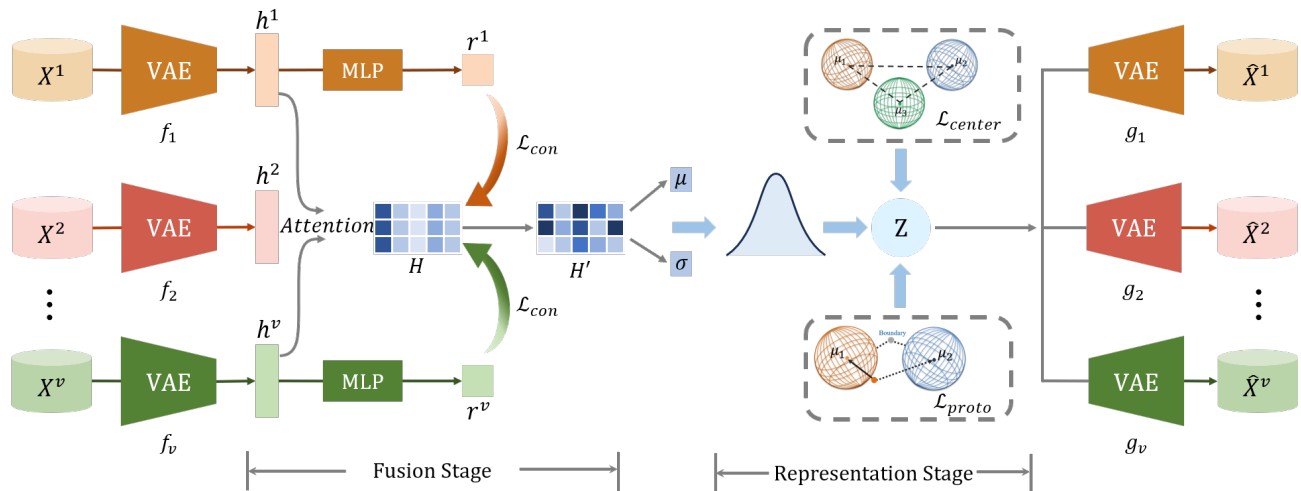

*Figure 1.* Dual-stage Contrastive Learning-enhanced Multi-view Variational Clustering (DCL-MVC) framework. Multi-view data are encoded into view-specific representations and processed through two contrastive learning stages. At the fusion stage, an attention-based fusion module constructs a unified representation, where instance-level contrastive learning enforces cross-view alignment and instance consistency. At the representation stage, boundary-aware contrastive learning is performed in the latent space, combining cluster-center contrast to enlarge inter-cluster margins with prototype contrast to refine ambiguous boundary assignments. The learned latent embedding is finally decoded to reconstruct each view.

approximate it. We further adopt a mean-field factorization and parameterize the variational posterior with two factors:

$$q(z, c \mid \{x^v\}_{v=1}^V) = q(z \mid \{x^v\}_{v=1}^V)\, q(c \mid \{x^v\}_{v=1}^V). \quad (7)$$

We parameterize the variational posterior using view-specific encoders. Specifically, the $v$-th view is mapped to a hidden representation $h^v$ through an encoder network $f_{\phi_v}(\cdot)$ parameterized by $\phi_v$, followed by a projection network that produces the Gaussian posterior parameters. Formally, we encode the $v$-th view as

$$h^v = f_{\phi_v}(x^v), \quad (8)$$

and use a view-specific projection head $p_{\psi_v}(\cdot)$ to obtain the mean and scale parameters

$$[\mu_v, \sigma_v] = p_{\psi_v}(h^v), \quad (9)$$

which defines the view-specific variational posterior

$$q_{\phi_v}(z^v \mid x^v) = \mathcal{N}(z^v \mid \mu_v, \sigma_v^2 \mathbf{I}). \quad (10)$$

Building on conventional view-fusion strategies (Xu et al., 2021), we introduce a cross-view attention module to adaptively fuse heterogeneous views. Unlike applying attention within each view separately, we calculate view importance in the shared latent space by learning attention weights over $\{h^v\}_{v=1}^V$; the fusion details are given in the next subsection.

### 3.4. Contrastive Fusion Stage

**View-level attention for multi-view fusion.** Within the VAE framework, we fuse multi-view information by

stacking the output of the view-specific encoder along the view dimension, yielding a view sequence $h = [h^1, \ldots, h^v, \ldots, h^V]^\top \in \mathbb{R}^{V \times d}$, where $h^v \in \mathbb{R}^{d_h}$ is the embedding of view $v$ and $d_h$ denotes the embedding dimensionality. To model the interactions among views, we apply a self-attention mechanism to $h$ and obtain an attention weight matrix $A \in \mathbb{R}^{V \times V}$. The attention weights are then computed as

$$A = \text{softmax}\left(\frac{QK^\top}{\sqrt{d_k}}\right), \quad (11)$$

where $d_k$ is the dimensionality of the key vectors and the softmax is applied row-by-row. The entry $A_{j,v}$ describes the attention weight of the $j$-th view to the $v$-th view.

Unlike conventional methods that directly use the attention-weighted outputs, we further average the attention weights over the view dimension to capture the relative importance of each view within the entire multi-view structure. Specifically, we define the global importance coefficient of the $v$-th view as

$$\alpha_v = \frac{1}{V} \sum_{j=1}^V A_{j,:}. \quad (12)$$

The final view-level fused representation is computed as

$$H = \sum_{v=1}^V \alpha_v h^v, \quad (13)$$

yielding an adaptively weighted aggregation of view-specific representations. The weights capture view-level

importance from a global perspective by modeling cross-view interactions in the shared latent space.

**Instance-level consensus alignment.** The linearly weighted fusion in Eq. (12)-(13) mainly captures view-level importance, but it does not resolve the instance-level semantic misalignment that can remain in the fused representation. As a result, embeddings of the same instance may still be inconsistent across views after fusion. Therefore, we further introduce consensus representation learning and instance-level contrastive constraints.

For each view $v$, we use a multi-layer perceptron (MLP) as a feature extraction module, which projects the latent representation $h^v$ to obtain the view-specific representation $r^v$. In the contrastive learning module, for each sample $i$, we treat $(r_i^v, H_i)$ as a positive pair and $\{(r_i^v, H_j)\}_{j \neq i}$ as negative pairs, where $H_i$ denotes the fused representation of sample $i$. This instance-level alignment encourages the fused representations to be more discriminative. Let $s(r_i^v, H_j)$ denote the cosine similarity.

For $v$-th view of $i$-th sample and the fused representation, the contrastive loss is defined as

$$\mathcal{L}_{\text{con}}^{(v,i)} = -\log \frac{\exp\left(s(r_i^v, H_i)/\tau\right)}{\sum_{j=1}^{N} \exp\left(s(r_i^v, H_j)/\tau\right)}, \quad (14)$$

where $\tau > 0$ is a temperature parameter. Although the weight of the $v$-th view $\alpha_v$ characterizes the global importance of each view, the agreement between the consensus features $r^v$ and the global features $H$ can vary between views. We therefore emphasize, at the instance level, views whose feature distributions are closer to the global representation. To this end, we adopt the maximum mean discrepancy (MMD) (Wu et al., 2024) to quantify the distributional discrepancy between $r^v$ and $H$, and use it to adaptively reweight the contribution of each view in the contrastive objective in Eq. (14).

Formally, we compute the squared MMD between $r^v$ and $H$ using a kernel function $k(\cdot, \cdot)$, denoted by $\mathcal{D}(r^v, H)$. We then convert it into a view-wise weight via softmax normalization

$$w_v = \text{softmax}\left(-\mathcal{D}(r^v, H)\right). \quad (15)$$

Accordingly, the overall instance-level contrastive loss is

$$\mathcal{L}_{\text{con}} = \sum_{v=1}^{V} w_v \frac{1}{N} \sum_{i=1}^{N} \mathcal{L}_{\text{con}}^{(v,i)}. \quad (16)$$

Minimizing Eq. (16) encourages the view-specific feature $r_i^v$ to align with the corresponding fused representation $H_i$, while pushing apart representations of different samples, thereby improving instance-level discriminability in the fused latent space.

After alignment of the view-level and the instance-level above, we denote the fused and aligned shared representation by $H'$. We then use a shared projection network $p_\psi(\cdot)$, as in Eq. (9), to map $H'$ to the Gaussian posterior parameters $(\mu, \sigma)$, thus obtaining the approximate posterior of the global latent variable:

$$q_\phi\left(z \mid \{x^v\}_{v=1}^V\right) = \mathcal{N}\left(z \mid \mu, \sigma^2 I\right). \quad (17)$$

We adopt the reparameterization trick to enable gradient backpropagation through sampling.

Following VaDE (Jiang et al., 2016), we compute the variational posterior over cluster assignments via the Bayes' rule based on a Monte Carlo sample $z^{(1)} \sim q_\phi\left(z \mid \{x^v\}_{v=1}^V\right)$:

$$q\left(c_k = 1 \mid \{x^v\}_{v=1}^V\right) = \frac{\pi_k \mathcal{N}\left(z^{(1)} \mid \mu_k, \sigma_k^2 I\right)}{\sum_{l=1}^K \pi_l \mathcal{N}\left(z^{(1)} \mid \mu_l, \sigma_l^2 I\right)}. \quad (18)$$

### 3.5. Latent Representation Stage

In the latent representation stage, we build on the contrastively fused representation to learn a more discriminative latent space for clustering. Each sample $x_i$ is associated with a variational posterior $\{q(c_k = 1 \mid x_i)\}_{k=1}^K$. For boundary samples, the posterior can be highly uncertain, weakening inter-cluster separability and introducing erroneous pseudo-labels. We tackle the boundary samples issue with two complementary latent-space contrastive mechanisms, namely cluster-center contrastive learning and prototype contrastive learning, to regularize uncertain representations and enhance inter-cluster discrimination.

We identify boundary samples using the margin between the top-1 and top-2 posterior probabilities. For each sample $x_i$, let $k_1$ and $k_2$ denote the indices of the most probable and the second most probable clusters, respectively. We regard $x_i$ as a boundary sample if

$$q(c_{k_1} = 1 \mid x_i) - q(c_{k_2} = 1 \mid x_i) < \theta, \quad (19)$$

where $\theta > 0$ controls the ambiguity level; otherwise, $x_i$ is treated as a high-confidence sample.

**Cluster-center contrastive learning.** Under the Gaussian mixture prior, $\mu_k$ is the mean of the $k$-th Gaussian component and naturally serves as the cluster center in the latent space. To reduce the overlap among different components and enlarge inter-cluster margins, we explicitly encourage these centers $\{\mu_k\}_{k=1}^K$ to be well separated by imposing a repulsive contrastive regularization. The cluster-center contrastive loss is defined as

$$\mathcal{L}_{\text{center}} = \frac{1}{K} \sum_{k=1}^K \log \sum_{l \neq k} \exp\left(s(\mu_k, \mu_l)/\tau\right), \quad (20)$$

This loss encourages different cluster centers to separate each other, leading to better separated Gaussian components and less overlap between posteriors of different clusters. In particular, a larger separation between centers makes the assignment of clusters for boundary samples more discriminative, leading to more confident posteriors $q(c_k = 1 \mid \{x^v\}_{v=1}^V)$.

**Prototype contrastive learning with curriculum.** While cluster-center contrastive learning enlarges inter-cluster margins, it doesnot explicitly refine instance assignments, especially for ambiguous samples near cluster boundaries when the cluster structure is still unstable. We therefore introduce prototype contrastive learning to encourage boundary samples to align with their assigned prototypes, by contrasting each sample representation $z_i$ against the set of cluster prototypes $\{\mu_k\}_{k=1}^K$. For each $z_i$, we compute its similarity-based matching probability to prototype $\mu_k$ as

$$p_{ik} = \frac{\exp\big(s(z_i, \mu_k)/\tau\big)}{\sum_{l=1}^K \exp\big(s(z_i, \mu_l)/\tau\big)}. \tag{21}$$

To avoid unreliable hard assignments at early training stages, we use the variational posterior $q(c_k = 1 \mid x_i)$ as a soft target and define the soft-target prototype contrastive loss as

$$\mathcal{L}_{\text{proto}}^{\text{soft}} = -\frac{1}{N} \sum_{i=1}^N \sum_{k=1}^K q(c_k = 1 \mid x_i) \log p_{ik}. \tag{22}$$

This objective encourages $z_i$ to align with prototypes in proportion to its posterior belief. As training progresses, $q(c_k \mid x_i)$ becomes sharper and approaches a one-hot assignment, yielding an implicit curriculum that guides boundary samples toward more confident cluster memberships.

As training proceeds, the posterior assignments for samples become sharper. Relying on soft targets may limit intra-cluster compactness. We adopt a confidence-aware curriculum that shifts from soft supervision to hard supervision. Let $k_i^*$ denote the most probable cluster index of sample $x_i$ under the posterior $q(c_k = 1 \mid x_i)$. In the intermediate stage, we apply a hard-label prototype contrastive loss to samples in $\bar{B}$ while keeping soft targets for boundary samples in $B$. Implementation details of boundary identification and the confidence-aware soft-to-hard curriculum are provided in Appendix A. The hard-label loss is defined as

$$\mathcal{L}_{\text{proto}}^{\text{hard}} = -\frac{1}{|\bar{B}|} \sum_{i \in \bar{B}} \log p_{ik_i^*}, \tag{23}$$

Boundary samples $i \in B$ still use the soft-target loss in Eq. (22). In the late stage, as posteriors become highly peaked, we further increase hard supervision by relaxing the boundary criterion, e.g., using $\theta/2$ in Eq. (19), so that only a few highly ambiguous samples remain under soft supervision. Overall, the training schedule gradually shifts

---

**Algorithm 1** Optimization of DCL-MVC

**Input:** Multi-view dataset $\{\{x_i^v\}_{v=1}^V\}_{i=1}^N$; number of clusters $K$; temperature $\tau$; boundary threshold $\theta$ (with curriculum schedule); trade-off parameters $\beta, \gamma$.
**Initialize:** $\{f_{\phi_v}, p_{\theta_v}\}_{v=1}^V$; attention module; consensus projection module; shared projection head; $\{\pi_k, \mu_k, \sigma_k^2\}_{k=1}^K$.
**while** not reaching the maximal epochs **do**
  1. Compute view-specific representations $\{h^v\}_{v=1}^V$ by Eq.(8).
  2. Obtain $\{(\mu_v, \sigma_v)\}_{v=1}^V$ by Eq.(9).
  3. Compute $\{\alpha_v\}_{v=1}^V$ and obtain $H$ by Eq.(11)–Eq.(13).
  4. Compute $\mathcal{L}_{con}$ and obtain $H'$ by Eq.(14)–Eq.(16).
  5. Compute $q_\phi(z \mid \{x^v\}_{v=1}^V)$ by Eq.(17).
  6. Compute $q(c \mid \{x^v\}_{v=1}^V)$ by Eq.(18).
  7. Compute $B, \bar{B}$ by Eq.(19) and $\mathcal{L}_{boundary}$ by Eq.(20)–Eq.(24).
  8. Update $\{\phi_v, \theta_v\}_{v=1}^V$, $\psi$, and $\{\pi_k, \mu_k, \sigma_k^2\}_{k=1}^K$ by minimizing Eq.(26).
**end while**
**Output:** $\hat{y}_i = \arg\max_k q(c_k \mid \{x_i^v\}_{v=1}^V)$.

---

from soft supervision during the early training phase, to a combination of soft and hard supervision, and finally to fully hard supervision.

Combining the two contrastive mechanisms, the objective of the boundary sample processing module is

$$\mathcal{L}_{\text{boundary}} = \mathcal{L}_{\text{center}} + \mathcal{L}_{\text{proto}}, \tag{24}$$

where $\mathcal{L}_{\text{proto}}$ denotes the stage-dependent prototype contrastive loss formed by combining the soft and hard terms.

### 3.6. Overall Objective

We learn the model by maximizing the evidence lower bound (ELBO) on multi-view data via variational inference. For the proposed multi-view Gaussian-mixture generative model and its inference network, the ELBO can be written as

$$
\begin{aligned}
&\mathcal{L}_{\text{ELBO}}(\{x^v\}_{v=1}^V) \\
&= \mathbb{E}_{q_\phi(z|\{x^v\}_{v=1}^V)} \left[ \sum_{v=1}^V \log p_\theta(x^v \mid z) \right] \\
&\quad - \mathbb{E}_{q_\phi(c|\{x^v\}_{v=1}^V)} \left[ D_{\text{KL}}\big(q_\phi(z \mid \{x^v\}_{v=1}^V) \,\|\, p(z \mid c)\big) \right] \\
&\quad - D_{\text{KL}}\big(q_\phi(c \mid \{x^v\}_{v=1}^V) \,\|\, p(c)\big).
\end{aligned}
\tag{25}
$$

On top of the ELBO, we incorporate an instance-level contrastive loss $\mathcal{L}_{\text{con}}$ from the feature fusion level and a boundary-aware contrastive loss $\mathcal{L}_{\text{boundary}}$ in the latent space. The final training objective is

$$\mathcal{L}_{\text{total}} = \mathcal{L}_{\text{ELBO}} + \beta \, \mathcal{L}_{\text{con}} + \gamma \, \mathcal{L}_{\text{boundary}}, \tag{26}$$

where $\beta$ and $\gamma$ balance the generative objective and the contrastive regularizers.

# 4. Experiment

## 4.1. Experimental Settings

**Datasets and Metrics.** In our experiments, we adopt six real-world datasets. Specifically, **Caltech7-5v** (Fei-Fei et al., 2004; Li et al., 2015) contains 1400 RGB images in 7 categories, and includes 5 views. **Scene-15** (Fei-Fei & Perona, 2005) contains 4485 scene images across 15 categories, each covering 3 views. **Prokaryotic** (Brbić et al., 2016), contains 551 prokaryotic species, each described by 3 views and categorized into 4 classes. **Cifar10**[1] is consisted of 10 categories of objects, contains 50000 samples and each item has 3 views. **HandWritten** (Nie et al., 2018) contains 2000 samples each of which is one of the handwritten digits(0-9). In the experiment, we use the PIX(240), FOU(76) and KAR(64) as 3 views. **MSRC-V1**[2] contains 210 images and 7 object classes with 5 views. The detailed statistics of these datasets are summarized in Table 2.

We evaluate the performance of multi-view clustering (MVC) methods using four widely adopted metrics: clustering accuracy (ACC), normalized mutual information (NMI), adjusted Rand index (ARI), and purity (PUR). For all these metrics, larger values indicate better clustering performance.

**Compared Methods.** To demonstrate the performance of our proposed DCL-MVC, we compare our method with the following state-of-the-art multi-view clustering approaches, i.e., DVIMVC(Xu et al., 2024), DCMVSC(Yu et al., 2024), MVP(Gao & Pu, 2025), DIMC(Xu et al., 2022), Multi-VAE(Xu et al., 2021) and DCMVC(Cui et al., 2024). Among them, DCMVSC can handle only two views. In our experiments, we provide the DCMVSC with the optimal two-view subset for each dataset to achieve its best performance. Detailed information on each compared method is provided in Appendix B.1.

## 4.2. Experimental Results and Analysis

The clustering performance on six datasets is reported in Table 1 and Table 3 using ACC, NMI, ARI and PUR. All competing methods are evaluated according to their recommended parameter settings and repeated 10 times. As shown in the tables, our method consistently achieves highly competitive results in most datasets. For example, on Prokaryotic dataset, our approach outperforms the strongest baseline by 11.69%, 2.57%, 12.9% and 12.55% in terms of ACC, NMI, ARI and PUR. Similar improvements are observed

---

[1]http://www.cs.toronto.edu/kriz/cifar.html

[2]https://www.microsoft.com/en-us/research/project/image-understanding/

on HandWritten and Caltech7-5v datasets, indicating that the proposed deep multi-view generative clustering model with a Gaussian-mixture prior and a shared latent variable learns a good representation while preserving cross-view consistency.

In addition, we compare our framework with a traditional view-specific attention variant, where attention is applied after each single-view embedding $h^v$ and then fused, while keeping the remaining modules unchanged, i.e. Pre-Attention. On Caltech7-5v and HandWritten datasets, our method yields clear gains of 21.57%, 13.38%, 24.10%, 10.51% and 24.15%, 16.20%, 25.22%, 22.60% on ACC, NMI, ARI and PUR, validating that fusion-then-self-attention better captures globally important view weights, rather than overemphasizing views that are only locally salient but not optimal for global clustering. We also compare the adopted single-sample posterior estimation with its multi-sample variant, and the results in Appendix B.2 show that comparable performance and supports the practicality of the single-sample design.

To further examine the robustness of DCL-MVC, we conduct additional experiments under noisy and missing-view settings on the Prokaryotic dataset. The detailed results are provided in Appendix B.3. DCL-MVC shows only limited degradation under increasing Gaussian noise, while its performance remains relatively stable under moderate missing rates and degrades under high missing rates. These results indicate that our method is reasonably robust to noisy views, whereas handling severe view missingness remains a meaningful direction for future work.

## 4.3. Complexity Analysis

Let $N$, $V$, and $K$ denote the numbers of samples, views, and clusters, respectively. The view-specific encoding and decoding modules cost $O(VN)$, and the self-attention fusion module costs $O(V^2N)$. The instance-level consensus alignment compares cross-sample relations and costs $O(VN^2)$. The posterior computation and boundary-aware learning cost $O(KN)$. In addition, the cluster-center contrastive learning costs $O(K^2)$, while the prototype contrastive learning costs $O(KN)$. Since $K, V \ll N$ in typical multi-view clustering scenarios, the overall complexity is dominated by the instance-level alignment term, i.e., $O(VN^2)$.

## 4.4. Ablation Study Analysis

To examine the contribution of each component, Table 4 reports ablation results on MSRC-V1. The results show that removing any component consistently degrades performance, verifying the contribution of each component.

Specifically, discarding the boundary sample processing module (w/o Boundary) yields a clear performance drop,

*Table 1.* Experiments on Caltech7-5v, Scene-15, Prokaryotic dataset. The best result is shown in **bold** and the second-best is underlined.

| Methods | Caltech7-5v | | | | Scene-15 | | | | Prokaryotic | | | |
|---|---|---|---|---|---|---|---|---|---|---|---|---|
| | ACC | NMI | ARI | PUR | ACC | NMI | ARI | PUR | ACC | NMI | ARI | PUR |
| BSV | 83.36 | 74.69 | 70.11 | 83.96 | 37.29 | 37.52 | 19.74 | 40.86 | 58.55 | 34.17 | 19.83 | 69.76 |
| DVIMVC(2024) | 88.53 | 81.22 | 78.61 | 88.57 | 47.94 | 46.71 | 20.82 | 50.90 | 51.47 | 33.91 | 22.83 | 72.47 |
| DCMVSC(2025) | 66.02 | 54.40 | 46.61 | 66.98 | 41.73 | 38.25 | 24.33 | 48.26 | 39.45 | 44.45 | 21.40 | 56.81 |
| MVP(2025) | 70.57 | 70.67 | 63.56 | 73.44 | 45.89 | 43.30 | 27.98 | 50.25 | 61.27 | 39.91 | 32.78 | 71.30 |
| DIMC(2022) | 82.93 | 74.11 | 70.78 | 84.10 | 44.69 | 40.96 | 25.65 | 47.19 | 45.67 | 20.35 | 10.45 | 62.67 |
| Multi-VAE(2021) | 62.22 | 52.60 | 44.07 | 63.89 | 31.36 | 30.15 | 15.24 | 35.65 | 33.11 | 14.75 | 16.32 | 47.63 |
| DCMVC(2024) | 79.29 | 76.38 | 69.87 | 83.14 | 47.98 | 45.74 | 30.80 | **53.29** | 47.55 | 26.80 | 13.49 | 65.88 |
| Pre-Attention | 70.57 | 71.67 | 60.08 | 70.57 | 39.26 | 41.70 | 22.33 | 42.82 | 57.17 | 33.68 | 20.38 | 71.14 |
| DCL-MVC | **92.14** | **85.05** | **84.18** | **92.14** | **49.77** | **51.01** | **32.13** | 52.02 | **72.96** | **47.02** | **45.68** | **83.85** |

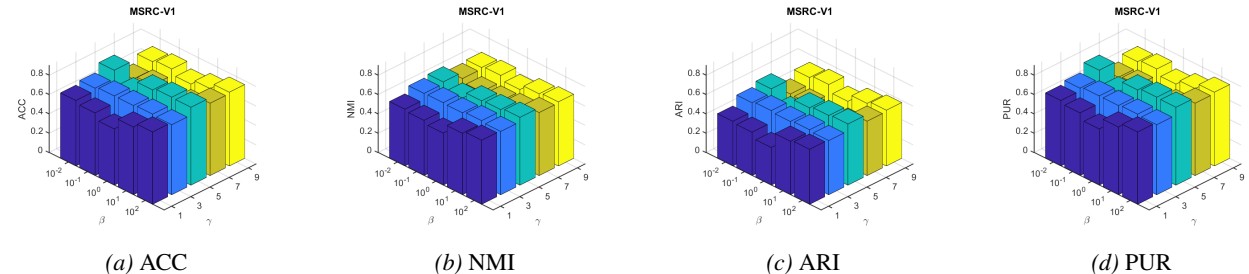

*(a)* ACC     *(b)* NMI     *(c)* ARI     *(d)* PUR

*Figure 2.* Parameter $(\beta, \gamma)$ sensitivity analysis on MSRC-V1.

*Table 2.* Statistics of the datasets.

| Dataset | Size | Views | Classes |
|---|---|---|---|
| Caltech7-5v | 1400 | 5 | 7 |
| Scene-15 | 4485 | 3 | 15 |
| Prokaryotic | 551 | 3 | 4 |
| Cifar10 | 50000 | 3 | 10 |
| HandWritten | 2000 | 3 | 10 |
| MSRC-V1 | 210 | 5 | 7 |

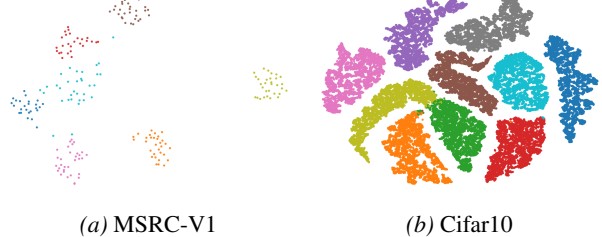

*(a)* MSRC-V1     *(b)* Cifar10

*Figure 3.* The visualization results on MSRC-V1 and Cifar10 dataset. Visualization results on other datasets are provided in Appendix B.6.

indicating that explicitly regularizing ambiguous boundary samples helps reduce cluster confusion and improve the clustering performance. In addition to quantitative ablation, we further provide an analysis in Appendix B.4 showing that the proposed boundary identification aligns better with a geometry-inspired ambiguity proxy, which supports the effectiveness and rationality of the boundary module.

Removing the instance-level consensus alignment (w/o Instance-level Fusion) causes more degradation, suggesting that instance-level cross-view agreement is crucial for learning a discriminative shared representation. Finally, the removal of the view-level self-attention fusion (w/o Self-attention) results in the performance decline, confirming that modeling global view importance in a unified latent space is critical for effective multi-view clustering.

We further remove the confidence-aware curriculum mechanism from the prototype contrastive learning objective. This confirms that the curriculum mechanism allows the model to gradually exploit more reliable samples and thus improves the effectiveness of prototype contrastive learning.

### 4.5. Parameter Sensitivity Analysis

In Eq. (26), there are two parameters $\beta$ and $\gamma$ to balance the strengths of the contrastive term and the boundary-related regularization, respectively. We conduct the parameter sensitivity analysis, and Fig. 2 reports the performance on MSRC-V1 with different values of $(\beta, \gamma)$. Results of the same sensitivity analysis on other datasets are provided in Appendix B.5. Overall, the performance changes only

*Table 3.* Experiments on Cifar10, HandWritten, MSRC-V1 dataset. The best result is shown in **bold** and the second-best is underlined.

| Methods | Cifar10 | | | | HandWritten | | | | MSRC-V1 | | | |
|---|---|---|---|---|---|---|---|---|---|---|---|---|
| | ACC | NMI | ARI | PUR | ACC | NMI | ARI | PUR | ACC | NMI | ARI | PUR |
| BSV | 89.32 | 78.55 | 77.83 | 89.32 | 75.50 | 74.25 | 65.09 | 78.36 | 66.90 | 56.43 | 46.65 | 67.48 |
| DVIMVC(2024) | 94.36 | 94.06 | 92.51 | 94.38 | 64.68 | 65.45 | 53.00 | 67.64 | 83.81 | 71.87 | 67.49 | 83.81 |
| DCMVSC(2025) | 96.40 | 92.81 | 91.53 | 96.40 | 83.85 | **81.38** | 74.50 | 83.85 | 82.28 | 75.78 | 69.25 | 71.67 |
| MVP(2025) | 90.57 | 81.10 | 83.71 | 94.89 | 83.89 | 78.82 | 74.66 | 83.46 | 83.86 | 70.41 | 69.89 | 87.86 |
| DIMC(2022) | 96.91 | 91.69 | 93.31 | 96.91 | 83.08 | 75.81 | 70.73 | 83.43 | 72.06 | 63.66 | 55.05 | 75.40 |
| Multi-VAE(2021) | 70.26 | 66.48 | 56.45 | 73.65 | 60.23 | 54.96 | 42.79 | 63.41 | 52.24 | 46.15 | 33.31 | 57.30 |
| DCMVC(2024) | 95.26 | 90.54 | 89.81 | 95.26 | 77.25 | 79.15 | 70.75 | 81.25 | 89.52 | 81.64 | 77.57 | 89.52 |
| Pre-Attention | 89.52 | 95.53 | 88.82 | 89.54 | 66.20 | 64.93 | 54.71 | 67.75 | 78.10 | 77.60 | 69.10 | 79.05 |
| DCL-MVC | **98.23** | **95.54** | **96.13** | **98.23** | **90.35** | 81.13 | **79.93** | **90.35** | **90.95** | **82.95** | **80.26** | **90.95** |

*Table 4.* Ablation experiment results on the MSRC-V1. The best in each column is shown in **bold**

| Methods | MSRC-V1 | | | |
|---|---|---|---|---|
| | ACC | NMI | ARI | PUR |
| w/o Boundary | 85.71 | 75.25 | 71.25 | 85.71 |
| w/o Instance-level | 80.47 | 72.30 | 65.31 | 80.47 |
| w/o Self-attention | 77.14 | 72.92 | 64.26 | 77.14 |
| w/o curriculum | 86.19 | 76.49 | 71.70 | 86.19 |
| DCL-MVC | **90.95** | **82.95** | **80.26** | **90.95** |

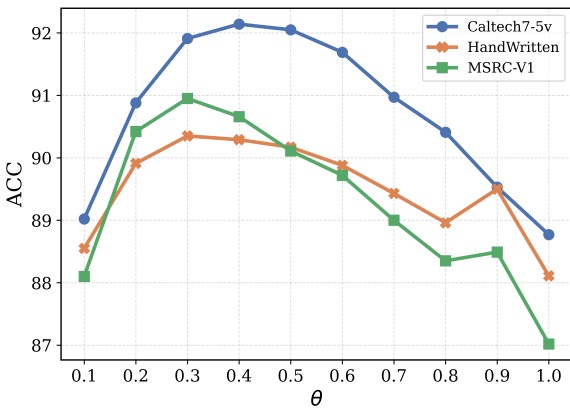

*Figure 4.* Parameter sensitivity analysis of the boundary threshold $\theta$ in terms of ACC on three representative datasets.

marginally on the tested grid, demonstrating that our method is robust to both hyperparameters. In particular, varying $\beta$ over several orders of magnitude results in a clear plateau for all metrics, with only a slight drop around $\beta \approx 10^0$, suggesting that overly strong contrastive pressure may mildly interfere with the generative clustering objective and the latent mixture structure. Meanwhile, increasing $\gamma$ tends to bring small yet consistent improvements across ACC, NMI, ARI and PUR, indicating that boundary regularization is beneficial for alleviating cluster ambiguity, although its effect remains moderate. In practice, we observe consistently strong results with a moderate $\beta$ and a relatively larger $\gamma$. These results verify that the proposed objective exhibits low hyper-parameter sensitivity and thus is easy to tune in practice.

We further analyze the sensitivity of the boundary threshold $\theta$. For clarity, Fig. 4 reports the ACC curves on three representative datasets, i.e., Caltech7-5v, HandWritten, and MSRC-V1. The results show that DCL-MVC remains generally stable under different choices of $\theta$, and better performance is mostly achieved within a moderate range of 0.3–0.6.

## 5. Conclusion

In this paper, we propose DCL-MVC, a dual-stage contrastive learning-enhanced multi-view variational clustering framework. To build a reliable fused representation and avoid being dominated by unreliable views, we design a contrastive fusion stage that performs fusion-then-attention in a unified latent space to learn global view-importance weights, together with instance-level consensus alignment to enhance cross-view consistency. To reduce boundary-sample misassignment caused by fusion-induced posterior ambiguity, we further introduce a latent representation stage to handle ambiguous posteriors. It increases inter-cluster margins through cluster-center repulsion and strengthens intra-cluster compactness through prototype contrast with a confidence-aware soft-to-hard curriculum. Extensive experiments on six real-world datasets demonstrate consistent improvements over strong baselines.

## Acknowledgements

This work is supported in part by the National Natural Science Foundation of China (No. 62276079), and the Special Funding Program of Shandong Taishan Scholars Project.

## Impact Statement

This paper presents work whose goal is to advance the field of Machine Learning. There are many potential societal consequences of our work, none which we feel must be specifically highlighted here.

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

We provide more details and results about our work in the appendix. Here are the contents:

## A. Boundary Samples and Confidence-aware Curriculum

We identify boundary samples based on the uncertainty of cluster assignment posterior and design a confidence-aware soft-to-hard curriculum accordingly. For each instance $x$, we first obtain one latent sample $z^{(1)}$ from the variational posterior $q_\phi(z \mid x)$ using the reparameterization trick. For efficiency and stability, we adopt a single Monte-Carlo sample in practice. Under the mixture-of-Gaussians prior, we compute the cluster-assignment posterior for $z^{(1)}$ and use it as an approximation of $q(c \mid x)$:

$$q(c = k \mid x) \approx p(c = k \mid z^{(1)}) = \frac{\pi_k \mathcal{N}(z^{(1)}; \mu_k, \Sigma_k)}{\sum_{j=1}^{K} \pi_j \mathcal{N}(z^{(1)}; \mu_j, \Sigma_j)}. \tag{27}$$

Given $q(c \mid x)$, we quantify the assignment confidence using a top1–top2 margin. Let $q_{(1)}(x)$ and $q_{(2)}(x)$ denote the largest and the second-largest probabilities in $q(c \mid x)$, respectively. The confidence score is defined as

$$s(x) = q_{(1)}(x) - q_{(2)}(x). \tag{28}$$

Intuitively, a small $s(x)$ indicates that the model is uncertain about the cluster membership of $x$, and such samples are more likely to lie near cluster boundaries. With a threshold $\theta$, we dynamically construct boundary and non-boundary sets within each mini-batch (rather than recomputing them over the whole dataset):

$$B = \{x \mid s(x) < \theta\}, \qquad \bar{B} = \{x \mid s(x) \geq \theta\}. \tag{29}$$

Based on the above partition, we employ a three-stage curriculum to progressively transition from soft to hard cluster assignments. Let the total number of training epochs be $T$ and the current epoch index be $t$. In the early stage ($t < 0.3T$), all samples use *soft* assignments, where $q(c \mid x)$ is directly used as soft targets to avoid introducing overly strong discrete supervision when assignments are still unstable. In the middle stage ($0.3T \leq t < 0.7T$), we apply *hard* assignments to confident samples in $\bar{B}$ while keeping soft targets for boundary samples in $B$. Specifically, the hard label is obtained by

$$\hat{c}(x) = \arg\max_k q(c = k \mid x). \tag{30}$$

In the late stage ($t \geq 0.7T$), we further expand the coverage of hard assignment by relaxing the confidence criterion: samples with $s(x) \geq 0.5\theta$ are treated as confident and assigned hard labels, and only highly uncertain samples continue to use soft targets. This strategy gradually shifts the learning focus from uncertainty-aware soft guidance to stronger discrete supervision, leading to stable and discriminative clustering.

Note that boundary identification and hard assignment involve discrete operations (thresholding and $\arg\max$), which do not create differentiable gradient paths through the discrete decisions; meanwhile, $q(c \mid x)$ is still used in differentiable terms (e.g., regularization and soft-target objectives) to maintain stable training while explicitly accounting for boundary ambiguity.

## B. More details about experimental settings and additional experiment results

### B.1. Datasets

We conduct the experiments on the following public datasets.

- **BSV**: BSV is a single-view approach, which conducts k-means clustering individually on each view, and chooses the best output as the final outcome.

- **DVIMVC**(Xu et al., 2024): Variational autoencoder-based Deep Variational Incomplete Multi-view Clustering (DVIMVC) uses view-specific encoders + Product-of-Experts aggregation, adds a coherence objective, and incorporates a Gaussian mixture prior to learn clustering-friendly shared representations for IMVC.

- **DCMVSC**(Yu et al., 2024): DCMVSC: Deep Cross-view Multi-view Subspace Clustering (DCMVSC) integrates contrastive learning, CS divergence-based loss, and block diagonalization constraints, unifying representation learning and clustering for enhanced multi-view clustering performance.

- **MVP**(Gao & Pu, 2025): Multi-View Permutation of Variational Auto-Encoders (MVP) excavates cross-view invariant relationships via latent-space correspondences, uses permutations for cross-view generation, and adds a cyclic permutation prior to boost consistency for incomplete multi-view data.

- **DIMC**(Xu et al., 2022): Imputation-free and fusion-free Deep Incomplete Multi-view Clustering (DIMC) builds per-view embedding models, maps complete data to high-dimensional space for linear separability, mines cross-view complementarity, and uses an EM-like strategy to boost clustering performance.

- **Multi-VAE**(Xu et al., 2021): VAE-based Multi-VAE disentangles visual representations via a view-common and view-peculiar variables, controlling mutual information to mine shared cluster factors for enhanced multi-view clustering.

- **DCMVC**(Cui et al., 2024): Deep Multi-view Clustering Network (DCMVC) adopts dual contrastive losses to learn clustering-friendly representations with inter-cluster separation and intra-cluster compactness.

To provide a more comprehensive evaluation, this appendix reports additional experimental results that are omitted from the main text due to space limitations. Specifically, we present (i) parameter sensitivity studies on the remaining five datasets, and (ii) qualitative visualizations on the remaining four datasets.

### B.2. Stability Analysis of Single-sample Posterior Estimation

Our method adopts the single-sample Monte Carlo approximation commonly used in variational models to balance trainability and computational cost. To examine whether this design introduces abnormal instability, we evaluate DCL-MVC on the Prokaryotic dataset over 10 independent runs and compare its mean and standard deviation with two strong baselines. As shown in Table 5, DCL-MVC maintains low standard deviations across all four metrics, suggesting that the boundary detection based on a single latent sample is stable in practice.

*Table 5.* Stability comparison on the Prokaryotic dataset over 10 independent runs.

| Method | ACC | NMI | ARI | PUR |
|--------|-----|-----|-----|-----|
| DCMVSC | 39.45±2.60 | 44.45±1.73 | 21.40±1.71 | 56.81±1.03 |
| MVP | 61.27±1.84 | 39.91±2.06 | 32.78±1.96 | 71.30±1.85 |
| DCL-MVC | 72.96±0.92 | 47.02±0.96 | 45.68±1.15 | 83.85±0.88 |

We also compare the computational cost of the single-sample and multi-sample variants. Given mini-batch size $B$, latent dimension $D$, cluster number $K$, and sample number $M$, the posterior computation costs $O(BKD)$ in the single-sample variant and $O(MBKD)$ in the multi-sample variant. Therefore, the additional cost of multi-sample estimation grows approximately linearly with $M$. Since the multi-sample variants do not yield clear performance gains, we use the single-sample version in the main experiments.

*Table 6.* Comparison between single-sample and multi-sample posterior estimation on the Prokaryotic dataset.

| $M$ | ACC | NMI | ARI | PUR |
|-----|-----|-----|-----|-----|
| 1 | 72.96 | 47.02 | 45.68 | 83.85 |
| 3 | 71.05 | 48.22 | 45.62 | 83.85 |
| 5 | 71.42 | 49.00 | 46.27 | 84.21 |

### B.3. Robustness Analysis on Noisy and Missing Views

We further conduct two direct robustness experiments on the Prokaryotic dataset, including noisy views and missing views.

**Noisy views.** We add i.i.d. zero-mean Gaussian noise $\mathcal{N}(0, \sigma^2)$ to the input features of each view and vary $\sigma$ to construct different noise levels. As shown in Table 7, when the noise standard deviation increases from 0.01 to 1.0, the performance shows only limited degradation, indicating that DCL-MVC is tolerant to feature perturbations in individual views.

*Table 7.* Robustness analysis under noisy views on the Prokaryotic dataset.

| $\sigma$ | 0.01 | 0.05 | 0.1 | 0.2 | 0.5 | 1.0 |
|---|---|---|---|---|---|---|
| ACC | 71.14 | 70.96 | 69.15 | 68.97 | 69.69 | 69.51 |
| NMI | 47.83 | 46.87 | 47.85 | 48.16 | 46.23 | 43.90 |
| ARI | 47.02 | 46.08 | 44.94 | 45.01 | 46.02 | 44.52 |
| PUR | 83.67 | 83.30 | 83.67 | 83.85 | 82.76 | 82.21 |

**Missing views.** We randomly generate view-missing masks under different missing rates. For each missing view, we fill the missing input with the corresponding feature mean vector computed from the full data of that view, and then feed the completed multi-view inputs into the encoder and fusion modules. As shown in Table 8, the model remains relatively stable at missing rates of 0.1 and 0.3, while the performance drops more noticeably when the missing rate increases to 0.5. This suggests that DCL-MVC is reasonably robust to moderate view missingness, but severe missingness remains challenging.

*Table 8.* Robustness analysis under missing views on the Prokaryotic dataset.

| Missing rate | 0.1 | 0.3 | 0.5 |
|---|---|---|---|
| ACC | 70.78 | 69.51 | 57.71 |
| NMI | 44.89 | 39.74 | 30.43 |
| ARI | 44.91 | 40.23 | 20.62 |
| PUR | 82.03 | 80.04 | 70.09 |

### B.4. Alignment between Posterior-based Boundary Identification and a Geometric Proxy

**Proxy ambiguous set and posterior boundary set.** For each sample, we compute a pair of scores $(g(x), s(x))$ and plot them as a scatter. Here, $s(x)$ denotes the posterior margin (top1–top2) used in our boundary identification (definitions are provided in the main text). To obtain a geometry-inspired proxy, we define $g(x) = d_2(x) - d_1(x)$, where $d_1(x)$ and $d_2(x)$ are the Euclidean distances from the sample representation $z(x)$ to the nearest and second-nearest prototypes, respectively. A smaller $g(x)$ indicates that the sample is geometrically closer to the cluster boundary.

We define the proxy ambiguous set $A$ as the bottom-$q$ quantile of $g(x)$:

$$A = \{x \mid g(x) \leq g_{\text{thr}}\}, \tag{31}$$

where $g_{\text{thr}}$ is the $q$-quantile threshold (shown as the vertical dashed line in the figures). Similarly, we define the posterior boundary set $B$ as the bottom-$q$ quantile of $s(x)$:

$$B = \{x \mid s(x) \leq \theta\}, \tag{32}$$

where $\theta$ is the $q$-quantile threshold of $s(x)$ (shown as the horizontal solid line). In our implementation (`theta_mode=quantile`), $\theta$ is *adaptively determined* from the empirical distribution of $s(x)$ under each setting (with / without the boundary module), which guarantees a fair comparison with the same boundary budget (i.e., the same proportion of selected samples).

In the scatter plots, the proxy ambiguous set $A$ (orange crosses) tends to concentrate on the *left* side because it only enforces small $g(x)$ (geometrically boundary-like), while the posterior boundary set $B$ (green circles) tends to concentrate on the *lower* side because it only enforces small $s(x)$ (posterior-uncertain). Therefore, $B$ appearing mainly in the lower-left region is expected: it corresponds to the most ambiguous samples that are *both* geometrically boundary-like and posterior-uncertain. Importantly, $A$ and $B$ are *not required* to fully coincide. Some samples may have small $g(x)$ but still yield a confident posterior assignment (large $s(x)$), i.e., *close to the boundary but still separable*; such samples need not be handled as boundary cases.

With the boundary module enabled, the posterior margin $s(x)$ typically exhibits a larger dynamic range: many in-cluster samples become more confidently assigned (larger $s(x)$), while genuinely ambiguous boundary samples remain at low $s(x)$. This yields a more spread-out distribution in the $(g(x), s(x))$ plane. In contrast, removing the boundary module often leads to a shrinkage of the posterior margin distribution, which is also reflected by the quantile-based threshold $\theta$ becoming very small under the w/o setting.

**Quantitative consistency.** To quantify the alignment between the two sets, we report (i) Jaccard similarity $\frac{|A \cap B|}{|A \cup B|}$ and (ii) Recall of $A$ covered by $B$, $\frac{|A \cap B|}{|A|}$, in the figure titles. On HandWritten, enabling the boundary module yields higher overlap and coverage, indicating that posterior uncertainty better matches the geometric boundary structure.

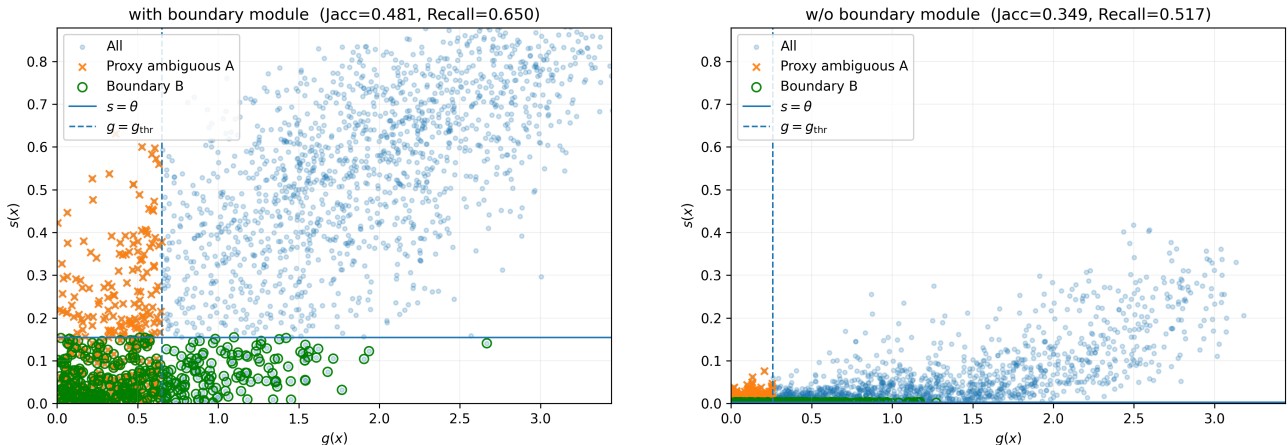

*Figure 5.* **HandWritten**: alignment between posterior-based boundary set $B$ (green circles, $s(x) \leq \theta$) and proxy ambiguous set $A$ (orange crosses, $g(x) \leq g_{\mathrm{thr}}$), where $\theta$ and $g_{\mathrm{thr}}$ are defined by the same bottom-$q$ quantile rule for fair comparison.

### B.5. Parameter Sensitivity on Additional Datasets

We further investigate the sensitivity of the proposed method with respect to key hyper-parameters on five additional datasets.

Overall, the performance curves exhibit consistent trends across datasets, indicating that the proposed method is not overly sensitive to moderate parameter variations. In most cases, the method maintains competitive performance within a reasonably wide range around the default setting, suggesting that the proposed method remains stable under different hyper-parameter choices.

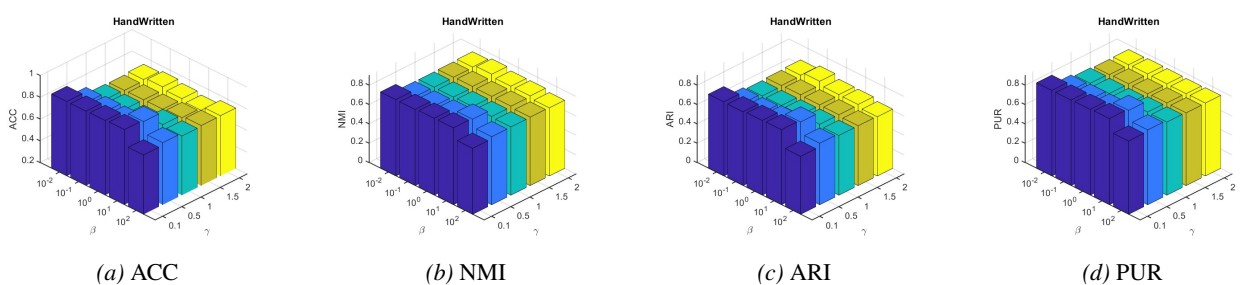

*(a)* ACC      *(b)* NMI      *(c)* ARI      *(d)* PUR

*Figure 6.* Parameter $(\beta, \gamma)$ sensitivity analysis on HandWritten.

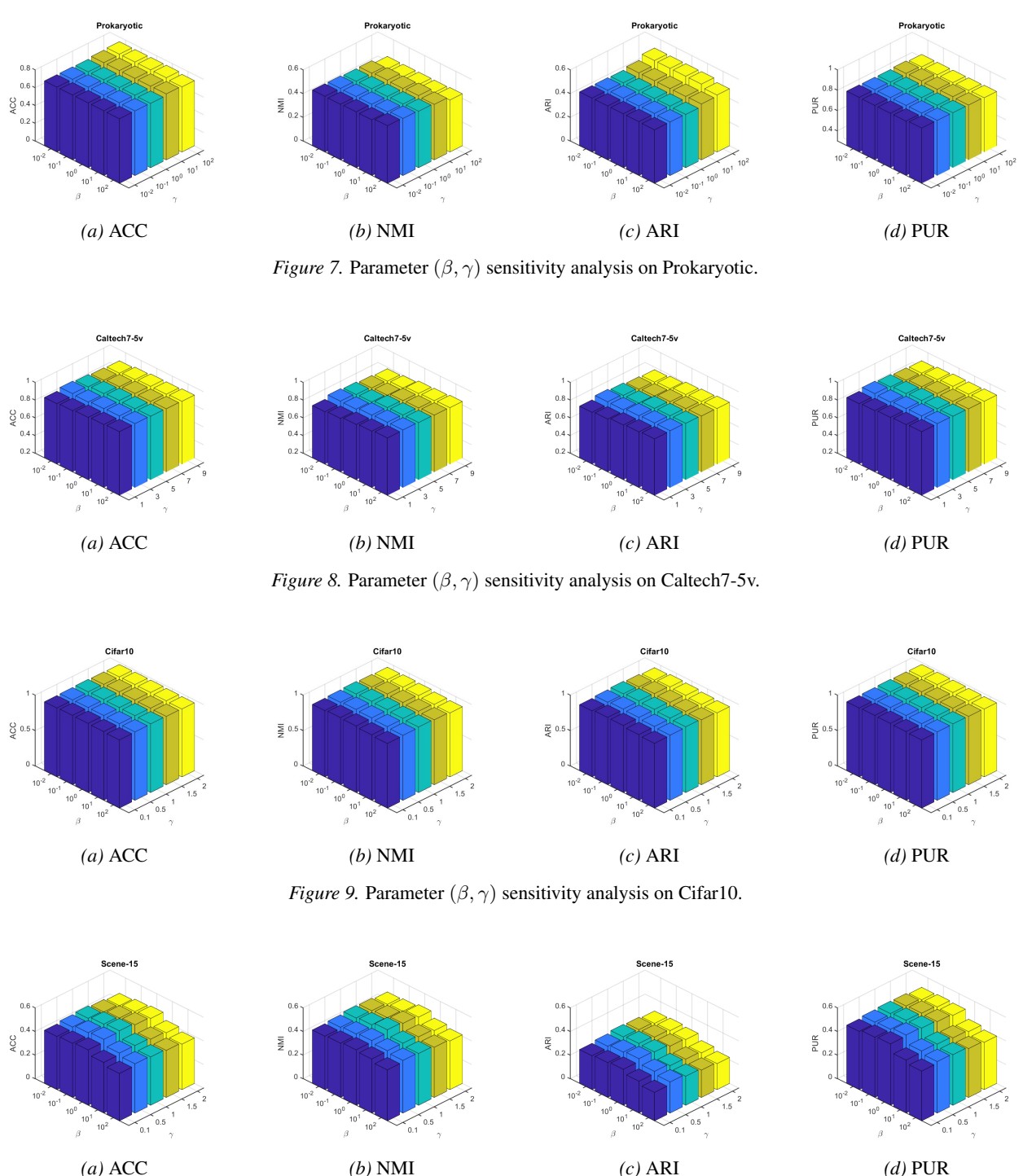

*(a)* ACC      *(b)* NMI      *(c)* ARI      *(d)* PUR

*Figure 7.* Parameter $(\beta, \gamma)$ sensitivity analysis on Prokaryotic.

*(a)* ACC      *(b)* NMI      *(c)* ARI      *(d)* PUR

*Figure 8.* Parameter $(\beta, \gamma)$ sensitivity analysis on Caltech7-5v.

*(a)* ACC      *(b)* NMI      *(c)* ARI      *(d)* PUR

*Figure 9.* Parameter $(\beta, \gamma)$ sensitivity analysis on Cifar10.

*(a)* ACC      *(b)* NMI      *(c)* ARI      *(d)* PUR

*Figure 10.* Parameter $(\beta, \gamma)$ sensitivity analysis on Scene-15.

## B.6. Visualization Results on Additional Datasets

To complement the quantitative results, we provide visualization results on four additional datasets. We visualize the learned representations (e.g., the fused latent representations) and the corresponding cluster assignments using the same protocol as in the main paper. Figure 11 shows that samples from different categories form well-separated clusters, while intra-class samples tend to be compact, which qualitatively supports the effectiveness of our view-consistent latent modeling and clustering-friendly representation learning.

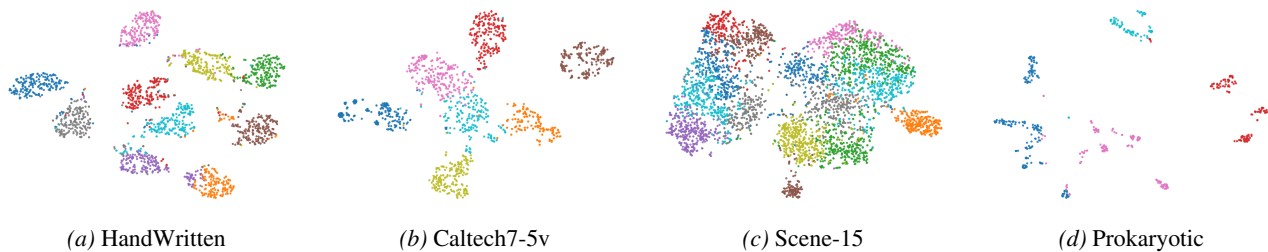

(a) HandWritten     (b) Caltech7-5v     (c) Scene-15     (d) Prokaryotic

*Figure 11.* The visualization results on other four datasets.

