# OpenReview forum: "Dual-stage Contrastive Learning-enhanced Multi-view Variational Clustering"
_ICML.cc/2026/Conference — ICML 2026 regular_

### Official Review · Reviewer_FUtg · 2026-03-05

**Soundness:** 3
**Presentation:** 4
**Significance:** 3
**Originality:** 3
**Overall Recommendation:** 4
**Confidence:** 3

**Summary:**

The paper presents a Multi-view Variational Clustering (MVC) method named DCL-MVC. MVC is a task which, given a set of data points which each contain multiple views (aka modalities), attempts to train a variational auto-encoder (VAE) in an unsupervised fashion to map multiple data modalities into a shared, probabilistic latent space, allowing it to assign clusters based on that underlying distribution. DCL-MVC proposes a fusion-then-attention strategy, where cross-view interaction is modeled as self-attention, from which view-level importance weights are learned to form a unified representation. DCL-MVC also proposes to introduce contrastive learning strategies to better separate the distributions between clusters, as well as improve robustness for ambiguous boundary samples. Experiments on several real-world datasets, as well as various statistical visualizations and hyperparameter sensitivity tests, effectively demonstrate the stability and robustness of the proposed method.

**Compliance With Llm Reviewing Policy:**

Affirmed.

**Final Justification:**

**Initial Assessment and Strengths/Weaknesses**: I found this paper to be well structured and technically strong. It introduces a novel and robust MVC framework (DCL-MVC) based on contrastive learning and showed its robustness to training hyperparameters via stability analyses on instance-level contrastive loss term weight $\beta$ and boundary-aware contrastive loss term weight $\gamma$. However, I had some concerns about whether it was also stable to the choice of the boundary threshold hyperparameter $\theta$, and what they meant by cross-level interactions.

**Rebuttal Evaluation**: The authors provided a clear and strong response that fully answered my concerns:

   • Robustness: The authors added a stability analysis for the boundary threshold hyperparameter $\theta$, showing that DCL-MVC was indeed insensitive to the same wide range of $\theta$ between 0.3 and 0.6 across various datasets. This makes a strong case for its applicability, as there is less need for hyperparameter tuning.

   • Visual Comparisons: Extra clustering visualizations on baseline methods being difficult to add immediately under the current rebuttal system is reasonable, and its absence is not critical. The authors’ promise to add additional clustering visual comparisons is sufficient improvement.

   • Clarification: The authors corrected cross-level interactions to cross-view interactions. This cleared up my confusion in that section.

**Final Recommendation**: The authors' reply fully addressed my concerns. The paper makes an important contribution by proposing a robust 2-stage contrastive-learning-based approach for variational MVC. As long as the promised visualization results are included in the final paper, I gladly maintain my **Weak Accept** rating.

**Key Questions For Authors:**

1. Figures 3 and 10 visualize the clustering results of the proposed method on various datasets, demonstrating that the proposed method indeed effectively maps multi-view data into well-separated and compact clusters. Can you perform similar visualizations on existing methods? I believe it would further highlight the effectiveness of the proposed method.

2. How did you determine the value of the threshold \theta? If it is a hyperparameter, did you perform stability analysis on this value? Clarification on this front would improve the soundness of your paper.

3. In page 4 column 2 lines 176-177, it says that your proposed adaptive weights model cross-level interactions in the shared latent space to capture view-level importance. Did you mean to say cross-view instead of cross-level? Further clarification of this part would improve the presentation of your paper.

**Limitations:**

Yes.

**Strengths And Weaknesses:**

- The paper is very well structured, the mathematics are concise and easy to understand, and the motivations and effects of each design element are well presented.
- The hyperparameter sensitivity analysis and its intuitive visualization effectively show the robustness to specific hyperparameter values, further supporting the robustness of the proposed method.
- The proposed design choices all address various aspects of ambiguities and instabilities throughout the training of Multi-view Variational Clustering, and their analyses offer novel insights into the field.

---

> ### Author Rebuttal · Authors · 2026-03-31
>
> **Q1**:
>
> **R1:** Thank you for the suggestion. As the rebuttal is not well suited for presenting additional visualization figures, we are unable to include more method visualizations here. However, we will add visualization comparisons with representative baseline methods in the revised manuscript to more comprehensively highlight the effectiveness of the proposed method.
>
> **Q2**:
>
> **R2:** The threshold $\theta$ is treated as a tunable hyperparameter in our method. We determined its default value through grid search over candidate values and selected a moderate setting that performed consistently well across datasets. We also have added a parameter sensitivity analysis for $\theta$. The results show that the model remains generally stable under different choices of $\theta$. Specifically, the better results on the six datasets are mostly achieved within a moderate range of **0.3–0.6**, while the performance declines to some extent when $\theta$ is set too small or too large. This indicates that the proposed boundary-aware mechanism is reasonably robust to the choice of $\theta$.
>
> $\theta$, ACC
>
> |             | 0.1   | 0.2   | 0.3   | 0.4   | 0.5   | 0.6   | 0.7   | 0.8   | 0.9   | 1.0   |
> | ----------- | ----- | ----- | ----- | ----- | ----- | ----- | ----- | ----- | ----- | ----- |
> | Caltech7-5v | 89.02 | 90.88 | 91.91 | 92.14 | 92.05 | 91.69 | 90.97 | 90.41 | 89.53 | 88.77 |
> | Scene-15    | 47.01 | 48.22 | 49.14 | 49.55 | 49.77 | 49.77 | 49.68 | 49.30 | 48.56 | 47.69 |
> | Prokaryotic | 69.57 | 71.02 | 72.44 | 72.88 | 72.96 | 72.81 | 72.33 | 71.90 | 72.01 | 70.22 |
> | Cifar10     | 96.93 | 97.66 | 98.11 | 98.20 | 98.23 | 98.19 | 97.95 | 98.01 | 97.32 | 96.44 |
> | HandWritten | 88.55 | 89.91 | 90.35 | 90.29 | 90.17 | 89.88 | 89.43 | 88.96 | 89.50 | 88.11 |
> | MSRC-V1     | 88.10 | 90.42 | 90.95 | 90.66 | 90.11 | 89.72 | 89.00 | 88.35 | 88.49 | 87.02 |
>
> **Q3**:
>
> **R3:** Thank you for pointing this out. Yes, we mean “cross-view” rather than “cross-level” here, since the adaptive weights $\alpha_v$ are learned to capture interactions among view-specific representations. We will correct this wording in the revised manuscript.

---

> > ### Author Rebuttal · Reviewer_FUtg · 2026-04-03
> >
> > I think that the authors has addressed my concerns in detail. In particular, the additional experiments on the threshold hyperparameter's stability further shows the robustness of the proposed method, showing its insensitivity to the hyperparameter values and increasing its practical applicability. I will maintain my previous score "Weak Accept".

---

### Official Review · Reviewer_jHHm · 2026-03-09

**Soundness:** 4
**Presentation:** 3
**Significance:** 3
**Originality:** 3
**Overall Recommendation:** 5
**Confidence:** 4

**Summary:**

This paper proposed a new multi-view clustering method under a variational framework to solve two key issues: unreliable cross-view fusion and ambiguous latent assignments, especially for boundary samples. The proposed new DCL-MVC is a dual-stage contrastive framework built on a VAE with a Gaussian mixture prior. At the fusion stage, the method uses a fusion-then-attention design with instance-level contrastive alignment to improve cross-view consistency. At the representation stage, it introduces cluster-center and prototype-based contrastive learning with a soft-to-hard curriculum to better handle uncertain samples. Experiments on six datasets show consistent improvements over recent baselines, supported by ablation studies, sensitivity analysis, and visualization results.

**Compliance With Llm Reviewing Policy:**

Affirmed.

**Key Questions For Authors:**

In the contribution section, the authors state that they “introduce a dual-stage formulation that decouples fusion reliability from posterior uncertainty.” Could the authors clarify in what sense this decoupling is achieved?
2. In the representation stage, prototype contrastive learning is combined with a curriculum schedule. How would performance change if the curriculum mechanism were removed?
3. Could the authors provide more quantitative evidence to show that the boundary-aware module indeed identifies meaningful boundary samples, rather than merely low-confidence ones?

**Limitations:**

yes

**Strengths And Weaknesses:**

Strength：
1. The paper is well motivated and identifies two concrete issues in multi-view variational clustering: unreliable fusion across heterogeneous views and ambiguous assignments for boundary samples in the latent space. The method addresses them separately through fusion-stage alignment and representation-stage refinement.
2. The method provides a relatively complete treatment of boundary samples by considering both their identification and subsequent refinement. In particular, it improves inter-cluster clarity through cluster-center separation, prototype-level contrastive learning, and a curriculum strategy.
3. The empirical evaluation is fairly comprehensive, including comparisons on six datasets, ablation studies, sensitivity analysis, and visualization results, which together strengthen the empirical support for the method.

Weakness：
1. The same notation variable "z"  used in Eq.(10) and Eq.(17), but it seems that they correspond to different posterior distributions. It would improve clarity to distinguish them with different symbols.
2. In Sec. 3.5, the Gaussian means {μ_k }are first introduced as cluster centers in Eq. (20), and then the same {μ_k }are used again as prototypes in Eq. (21)–(23). Although this is mathematically reasonable, the transition is somewhat abrupt in the current presentation and may leave readers wondering whether “cluster centers” and “prototypes” are exactly the same objects or play different roles.
3. The related-work discussion could better position the paper with respect to prior efforts on ambiguous-sample handling or uncertainty-aware clustering. The introduction explicitly motivates the method by emphasizing boundary samples with uncertain posteriors, but Sec. 2 mainly reviews deep multi-view clustering and VAE-based clustering at a broad level, without clearly isolating prior strategies that address ambiguous assignments or posterior uncertainty more directly.

---

> ### Author Rebuttal · Authors · 2026-03-31
>
> **W1:**
>
> **R1:** Thank you for pointing this out. We indeed used the same symbol $z$ in Eq. (10) and Eq. (17) to denote different meanings, which may cause confusion. Specifically, Eq. (10) refers to the view-specific latent posterior produced by each view-specific encoder, whereas Eq. (17) refers to the global latent posterior inferred from the fused representation. In the revised manuscript, we will rewrite the variable in Eq. (10) as $z^v$ and keep $z$ in Eq. (17) for the shared global latent variable.
>
> **W2:**
>
> **R2:** Thank you for pointing this out. In our method, the “cluster centers” and the “prototypes” actually refer to the same set of variables, namely the Gaussian means $\\{\mu_k\\}_{k=1}^K$ of the mixture prior. The difference is only in their roles in different parts, in Eq. (20), $\mu_k$ are emphasized as cluster centers for enlarging inter-cluster separation, while in Eq. (21)–(23), the same $\mu_k$ are used as prototypes for sample-to-prototype contrastive learning. We will make it explicit in the revised version.
>
> **W3:**
>
> **R3:** Thank you for this suggestion. In the revised manuscript, we will add a dedicated discussion on this aspect and include representative references [1] - [5] to better position our method with respect to prior work on uncertain assignments.
>
> [1] Xie, J., Girshick, R., and Farhadi, A. Unsupervised deep embedding for clustering analysis. In International conference on machine learning, pp. 478–487. PMLR, 2016.
>
> [2] Van Gansbeke, W., Vandenhende, S., Georgoulis, S., Proesmans, M., and Van Gool, L. Scan: Learning to classify images without labels. In European conference on computer vision, pp. 268–285. Springer, 2020.
>
> [3] Niu, C., Shan, H., and Wang, G. Spice: Semantic pseudo-labeling for image clustering. IEEE Transactions on Image Processing, 31:7264–7278, 2022.
>
> [4] Huang, Z., Chen, J., Zhang, J., and Shan, H. Learning representation for clustering via prototype scattering and positive sampling. IEEE Transactions on Pattern Analysis and Machine Intelligence, 45(6):7509–7524, 2022.
>
> [5] Yu, C., Shi, Y., and Wang, J. Contextually affinitive neighborhood refinery for deep clustering. Advances in Neural Information Processing Systems, 36:5778–5790, 2023.
>
> **Q1:**
>
> **R4:** The “Decouple” does not mean that fusion reliability and posterior uncertainty are statistically independent. Instead, it means that these two related issues are handled by two different stages in the model design. The first stage improves fusion reliability before posterior inference, while the second stage refines boundary uncertainty after the posterior is formed. Thus, “decouple” refers to a functional separation in optimization and problem handling, rather than an assumption of inherent independence. We will clarify this wording in the revised manuscript.
>
> **Q2:**
>
> **R5:** Following the reviewer’s suggestion, we added an ablation experiment on the **MSRC-V1** dataset by removing the curriculum mechanism. The results show that the model performance drops noticeably. This suggests that the curriculum mechanism is important for making prototype contrastive learning effective.
>
> |                | ACC       | NMI       | ARI       | PUR       |
> | -------------- | --------- | --------- | --------- | --------- |
> | w/o curriculum | 86.19     | 76.49     | 71.70     | 86.19     |
> | DCL-MVC   | **90.95** | **82.95** | **80.26** | **90.95** |
>
> **Q3:**
>
> **R6:** We have included related quantitative evidence in Appendix B.2, although it was not sufficiently emphasized in the main text. Specifically, we construct a geometry-based ambiguity proxy $g(x)$ independent of the posterior margin, and compare its ambiguous set with the posterior-based boundary set using Jaccard similarity and Recall. The results show that our identified boundary samples align well with the geometric boundary structure rather than being merely low-confidence ones. Moreover, this alignment becomes stronger when the boundary-aware module is enabled, further supporting the effectiveness of the proposed boundary identification strategy.

---

### Official Review · Reviewer_NuiR · 2026-03-11

**Soundness:** 3
**Presentation:** 3
**Significance:** 3
**Originality:** 4
**Overall Recommendation:** 5
**Confidence:** 4

**Summary:**

The paper proposes DCL-MVC, a dual-stage contrastive learning enhanced framework for multi-view variational clustering. The main motivation is that noisy and heterogeneous views can make fused representations unreliable, which in turn leads to overlapping posteriors and unstable assignments for boundary samples. To address this, the method uses a Gaussian-mixture VAE backbone with two contrastive stages. The first stage focuses on fusion, it models cross-view interactions through fusion-then-attention and improves view consistency via instance-level alignment. The second stage focuses on representation refinement, it enhances cluster separation and refines boundary samples through cluster-center regularization, prototype-based contrastive learning, and a confidence-aware soft-to-hard curriculum. Experiments on six real-world datasets show consistent gains over competitive baselines, with additional evidence from ablations and robustness analysis.

**Compliance With Llm Reviewing Policy:**

Affirmed.

**Final Justification:**

The authors have addressed my concerns after I reviewed their rebuttal.

**Key Questions For Authors:**

1. When comparing against DCMVSC, the manuscript states that the method is evaluated “with the optimal two-view subset for each dataset to achieve its best performance.” Could the authors clarify how the two views were selected for each dataset?
2. In Eq.(6), the authors assume conditional independence of views given the shared latent variable \mathrm{z}. Under this assumption, could the authors clarify how the subsequent cross-view fusion in the encoder is conceptually justified?
3. What is the motivation for adopting a dual-stage design? Would it be possible to replace the two-stage formulation with a unified latent-level contrastive objective?

**Limitations:**

yes

**Strengths And Weaknesses:**

Strength：
1. The dual-stage design is a notable strength. Instead of applying a generic contrastive loss, the paper uses different contrastive objectives at different stages to improve fusion quality and latent cluster discrimination respectively.

2. The fusion module is more carefully designed than simple view aggregation schemes, as it models cross-view interactions and combines view-level weighting with instance-level alignment.

3. The boundary-aware design is one of the more distinctive parts of the paper. By explicitly focusing on uncertain samples and refining them with prototype-based contrastive learning under a curriculum schedule, the method goes beyond uniform clustering objectives.

Weakness：
1. Eq.(10) appears potentially redundant, as the manuscript does not clearly utilize the view-specific latent variables in subsequent inference or generation steps. The role of these per-view posteriors should be clarified.

2. The notation for the cluster posterior is somewhat inconsistent across sections: Eq. (18) defines q($c_k=1\mid{x^v}_{v=1}^V$), while Sec. 3.5 later switches to q($c_k=1\mid x_i$) without explicitly clarifying whether this is merely shorthand for the multi-view observation of sample i. This may cause mild confusion about the conditioning variable.

3. The experimental section does not clearly specify the number of runs used for reporting results. It would be helpful to report the mean and standard deviation over multiple trials to evaluate statistical stability.

---

> ### Author Rebuttal · Authors · 2026-03-31
>
> **W1:**
>
> **R1:** Eq. (10) is not redundant. Its role is to describe the local variational posterior produced by each view encoder. Although the subsequent inference and generation steps do not directly use these view-specific posteriors, Eq. (10) is still a necessary part of the complete inference pipeline. It is introduced to explicitly present the process of “single-view encoding → cross-view fusion → global posterior inference.” We will clarify this point in the revised manuscript.
>
> **W2:**
>
> **R2:** Thank you for the suggestion. These two notations refer to the same quantity. In Sec. 3.5, $x_i$ is simply shorthand for the complete multi-view observation of the $i$-th sample, i.e., $\\{x^v_i\\}_{v=1}^V$. We used this notation only for brevity in the boundary identification step. We will clarify and unify the notation in the revised manuscript.
>
> **W3:**
>
> **R3:** The results of all methods on each dataset are obtained from 10 independent runs, the mean and standard deviation have been computed. Following the reviewer’s suggestion, we further report the mean and standard deviation of our method and representative baselines on the **Prokaryotic** dataset in the rebuttal to provide a more direct evaluation of statistical stability. The experimental results show that the proposed method not only achieves strong average performance across multiple datasets, but also exhibits relatively small standard deviations overall, indicating good stability.
> |Method   | ACC        | NMI        | ARI        | PUR        |
> | ----------- | ---------- | ---------- | ---------- | ---------- |
> | **DCL-MVC** | 72.96±0.92 | 47.02±0.96 | 45.68±1.15 | 83.85±0.88 |
> | DCMVSC      | 39.45±2.60 | 44.45±1.73 | 21.40±1.71 | 56.81±1.03 |
> | MVP         | 61.27±1.84 | 39.91±2.06 | 32.78±1.96 | 71.30±1.85 |
> | DVIMVC      | 51.47±2.15 | 33.91±2.42 | 22.83±2.93 | 72.47±2.32 |
>
> **Q1**:
>
> **R4:** Since DCMVSC can only handle two-view inputs, we enumerated all possible two-view combinations on each dataset and ran DCMVSC on each of them. We then reported the best-performing result among these combinations. This was done to reflect the best possible performance of DCMVSC and to ensure that the comparison is as comprehensive and fair as possible.
>
> **Q2**:
>
> **R5:** The conditional independence assumption in Eq. (6) is imposed at the generation stage. Its purpose is to specify that, in the generative model, once the shared latent variable $z$ is given, each view is independently generated by its own decoder. In contrast, the cross-view fusion is introduced in the variational inference stage: we first obtain view-specific features $h^v$, and then use self-attention and instance-level alignment to construct a fused representation before predicting the global posterior $q_{\phi}(z|{x^v})$. In short, the conditional independence in Eq. (6) is used to constrain the generative distribution, whereas cross-view fusion is an inference-side mechanism introduced to better estimate the approximate posterior. They operate at different levels and are therefore not contradictory.
>
> **Q3**:
>
> **R6:** The motivation for adopting a dual-stage design is that our method aims to address two issues at different levels: (1) unreliable cross-view fusion before posterior inference, and (2) uncertain assignment of boundary samples after the posterior is formed. Accordingly, the two stages are designed to improve fusion reliability and refine boundary assignment, respectively.
>
> In principle, it is possible to replace the dual-stage design with a unified latent-level contrastive objective. However, such an objective would typically be imposed only after fusion, and would therefore mainly act as a latent-space correction rather than explicitly improving fusion quality before posterior inference. As a result, fusion bias may already propagate into posterior estimation and later appear as boundary uncertainty. This would essentially bring the problem back to the two issues we aim to address.

---

> > ### Author Rebuttal · Reviewer_NuiR · 2026-04-02
> >
> > The authors address my concerns.

---

### Official Review · Reviewer_KJ8C · 2026-03-13

**Soundness:** 3
**Presentation:** 3
**Significance:** 3
**Originality:** 3
**Overall Recommendation:** 4
**Confidence:** 4

**Summary:**

This paper studies multi-view clustering and proposes a two-stage framework. The first stage is mainly for better view fusion, using attention and instance-level contrastive alignment. The second stage then tries to handle boundary samples that are more ambiguous, by adding a boundary-aware contrastive learning step in the latent space. The overall idea is that noisy or heterogeneous views can hurt fusion quality and eventually make clustering worse. The method is based on a VAE with a Gaussian mixture prior, and experiments are reported on six datasets.

**Compliance With Llm Reviewing Policy:**

Affirmed.

**Key Questions For Authors:**

1. Is the boundary detection really based on only one latent sample? If yes, how stable is this in practice, and did the authors compare it with a multi-sample version?
2. Can the authors provide more direct robustness experiments, for example with noisy views, missing views, or conflicting views?
3. How much extra cost does the full method add compared to the base model?

**Limitations:**

The paper does not explicitly discuss its limitations. It includes a brief impact statement, but this does not provide a meaningful discussion of assumptions, failure cases, or potential risks.

**Strengths And Weaknesses:**

**[Strength]**

1. The paper tackles a meaningful problem, since unreliable view fusion and ambiguous boundary samples are both practical issues in multi-view clustering.
2. The method combines several ideas in a way that mostly makes sense, and the two-stage design is easy enough to follow at a high level.
3. The results are generally solid across the six datasets, and the ablation gives at least some support that the main components matter.


**[Weakness]**

1. The boundary detection part does not feel fully stable to me. It looks like the paper uses the posterior from only one sampled latent variable, and then checks the gap between the top two cluster probabilities. For samples near the boundary, this could be quite noisy.
2. The paper talks a lot about robustness to noisy or heterogeneous views, but the experiments do not seem to test this claim directly. Right now the evidence is mostly standard benchmark results, ablation, and parameter sensitivity.
3. The method is also fairly heavy, with several added modules on top of the base model, but there is no discussion of runtime, memory, or scalability.

---

> ### Author Rebuttal · Authors · 2026-03-31
>
> **W1 & Q1**:
>
> **R1:** Yes. We use the single-sample Monte Carlo approximation commonly adopted in variational models to balance trainability and computational cost. We further evaluated our method on the **Prokaryotic** dataset over 10 independent runs, and compared its mean and standard deviation with those of the baselines. The results show that the standard deviation of our method remains low across all metrics, without exhibiting abnormal instability, indicating that the boundary detection based on a single latent sample is stable in practice.
> |Method   | ACC        | NMI        | ARI        | PUR        |
> | ----------- | ---------- | ---------- | ---------- | ---------- |
> | **DCL-MVC** | 72.96±0.92 | 47.02±0.96 | 45.68±1.15 | 83.85±0.88 |
> | DCMVSC      | 39.45±2.60 | 44.45±1.73 | 21.40±1.71 | 56.81±1.03 |
> | MVP         | 61.27±1.84 | 39.91±2.06 | 32.78±1.96 | 71.30±1.85 |
>
> Following the reviewer’s suggestion, we also implemented a multi-sample version by averaging cluster posteriors from multiple latent samples. The experimental results show that the multi-sample version(M=1,3,5) and the original single-sample version achieve very similar performance. We further compared the complexity of the single-sample and multi-sample variants. With mini-batch size $B$, latent dimension $D$, cluster number $K$, and sample number $M$, the posterior computation costs $O(BKD)$ in the single-sample version and $O(MBKD)$ in the multi-sample version. Hence, the extra cost grows approximately linearly with $M$. Since the multi-sample version does not show clear performance gains, we adopt the single-sample design as a practical trade-off between efficiency and effectiveness, and will clarify this in the revised paper.
> | M    | ACC   | NMI   | ARI   | PUR   |
> | ---- | ----- | ----- | ----- | ----- |
> | 1    | 72.96 | 47.02 | 45.68 | 83.85 |
> | 3    | 71.05 | 48.22 | 45.62 | 83.85 |
> | 5    | 71.42 | 49.00 | 46.27 | 84.21 |
>
> **W2 & Q2**:
>
> **R2:** We further supplemented two more direct robustness experiments on the **Prokaryotic** dataset: noisy views and missing views.
> (1) Noisy views. We added i.i.d. zero-mean Gaussian noise, $N(0,\sigma^2)$, to the input features of each view, and varied $\sigma$ to construct different noise levels. The results show that when the noise standard deviation increases from 0.01 to 1.0, the model performance exhibits only limited degradation, indicating that the proposed method has good tolerance to feature perturbations in individual views.
> | noise_std | ACC   | NMI   | ARI   | PUR   |
> | --------- | ----- | ----- | ----- | ----- |
> | 0.01      | 71.14 | 47.83 | 47.02 | 83.67 |
> | 0.1       | 69.15 | 47.85 | 44.94 | 83.67 |
> | 0.2       | 68.97 | 48.16 | 45.01 | 83.85 |
> | 0.5       | 69.69 | 46.23 | 46.02 | 82.76 |
> | 1.0       | 69.51 | 43.90 | 44.52 | 82.21 |
>
> (2) Missing views. We randomly generated view-missing masks under different missing rates, and filled each missing input with the corresponding feature mean vector computed from the full data of that view. The filled views were then fed into the encoder and fusion modules. The results show that the model remains relatively stable at missing rates of 0.1 and 0.3. When the missing rate further increases to 0.5, the performance drops more noticeably.
> | Missing rate | ACC   | NMI   | ARI   | PUR   |
> | ------------ | ----- | ----- | ----- | ----- |
> | 0.1          | 70.78 | 44.89 | 44.91 | 82.03 |
> | 0.3          | 69.51 | 39.74 | 40.23 | 80.04 |
> | 0.5          | 57.71 | 30.43 | 20.62 | 70.09 |
>
> Overall, these results indicate that the proposed method is reasonably robust to noisy views, while its performance degrades under high missing rate conditions, which we regard as an important direction for future work.
>
> **W3 & Q3**:
>
> **R3:** Let $N$, $V$, and $K$ denote the numbers of samples, views, and clusters. The full model has complexity  $O(VN)$ for the view-specific encoding and decoding, $O(V^{2}N)$for self-attention fusion, $O(VN^{2})$ for instance-level alignment, and $O(KN)$ for posterior computation and boundary-aware contrastive learning. The time complexity of the cluster-center contrastive learning is $O(K^{2})$, and the time complexity of the prototype contrastive learning is $O(KN)$. Since $K,V \ll N$, the total complexity is dominated by $O(VN^{2})$. We also regard the base model as the generative multi-view VAE backbone with the GMM prior, but without the self-attention fusion, instance-level consensus alignment, or boundary-aware contrastive modules. Its main time complexity is $O(VN + KN)$. Therefore, compared with the base model, the extra cost is mainly dominated by the instance-level alignment term $O(VN^{2})$.

---

> > ### Author Rebuttal · Reviewer_KJ8C · 2026-04-02
> >
> > I thank the authors for the detailed rebuttal. As my primary concerns have been adequately addressed, I will maintain my positive score.

---

### Decision · Program_Chairs · 2026-04-30

**Decision:**

Accept (regular)

**Comment:**

This paper proposes a dual-stage contrastive learning framework for multi-view variational clustering, addressing unreliable view fusion and ambiguous boundary samples through stage-specific contrastive objectives and a boundary-aware refinement mechanism. While reviewers agreed that the paper tackles a meaningful problem with a well-designed and empirically strong method, they also raised several concerns about evaluation depth, methodological robustness, and clarity. However, these concerns were largely addressed during rebuttal, leading to a clear consensus in favor of acceptance.